# Corporate Social Responsibility and Community Legitimacy: Colombian Caribbean Insights

María Cristina Bustillo-Castillejo [1] , Rosario Pérez-Morote [2,*] and Ángela González-Moreno [2]

1   Economic, Administrative and Accounting Sciences Faculty, University Libre,
    Cartagena de Indias 130001, Colombia; mariac.bustilloc@unilibre.edu.co
2   Management and Administration Department, University of Castilla-La Mancha, 02071 Albacete, Spain;
    angela.gonzalez@uclm.es
*   Correspondence: rosario.pmorote@uclm.es

**Abstract:** The success of companies and the recognition by the community in which they are inserted depends on the confidence that the company generates in this community and the approach to local development formulated by the community. In this sense, the impacts of CSR and the recognition of the company as an important agent within the community forge the reputation of the company in terms of its management and interrelations with the community. To analyze the factors that influence the recognition and legitimacy of companies by communities, this paper analyzes the communities' perception of territorial development and the impacts of CSR activities agreed in the social licenses in the context of Law 21/1991 on Prior Consultation in the Colombian Caribbean. Communities value investment in training and education from primary school to professional training, as well as income-generating practices. They also value respect for their culture, race, customs, and environmental wealth. CSR actions in healthcare do not provide greater legitimacy to the company. The same is the case with actions aimed to improve the relationship between suppliers and companies, as well as to strengthen the leadership of the community.

**Keywords:** corporate social responsibility; business legitimacy; recognition; prior consultation law

## 1. Introduction

Organizations are not individual entities with autonomy to decide their acts but rather they should be understood as actors in an environment, which defines them, while they also define certain aspects of that same environment [1]. In this sense, an organization has a set of rights and duties within a framework that regulates both the scope of its actions and the way it achieves its goals. Firms attempt to find a balance between expanding and implementing their projects and local development that guarantees the preservation of the culture, heritage, and identity of the territory in which they are located, its environmental diversity, and economic inclusion, which often means they find themselves involved in complex processes of relationships and participation.

The paradigm of sustainable development [2], which defends the ability to meet the needs of the present without compromising the ability of future generations to meet their own needs [3], has largely been neglected by business activity, despite it providing a useful perspective from which to conceptualize corporate social responsibility (CSR) [4–6].

The literature on guidelines for CSR actions has grown in recent years [7–10] and highlights the diversity in firms with CSR policies [11]. Authors have also identified the level of development of countries where CSR is implemented as an explanatory variable [12]. The work by Cuervo [13] is of interest here as it explains how investment in CSR evolves according to countries' level of development, considering the changes in the shortcomings of infrastructures and the cost of negative externalities. In less developed countries, CSR is a large, undiversified investment that focuses mainly on the social dimension to compensate for basic infrastructure deficits. In more advanced nations, the investment is less intense,

with new types of CSR directed at actions in the economic and environmental spheres. In developed countries, the diversity of actions is greatest, ranging across the three pillars of CSR: Social, economic, and environmental [13].

In Latin America, as a question of necessity and expectation, emphasis is placed on working conditions, the employment of more disadvantaged groups, philanthropy, and community development, including health and education, although these activities often have a reduced impact and are more a demonstration of solidarity with certain communities in response to the lack of support from governments, especially local ones. The true essence of CSR is distorted and adapted to the environment within which a firm operates [14,15].

In Colombia, many enterprises, whose processes of expansion and development are based on natural resources, the pillar of the Colombian economy, have to deal with environmental and social limitations, due to the State's inability to meet the needs and fundamental rights of its population, either through neglect or problems of corruption. Thus, private companies surrender their roles as generators of wealth and devote themselves to solving, mitigating, or contributing to the development of communities in questions of health, education, and income by means of productive inclusion as employers and suppliers of vulnerable populations, through solidarity and other practices related to fundamental rights.

The present work focuses on the impact of the CSR actions of firms that wish to launch a project in the Colombian Caribbean, specifically in the city of Cartagena de Indias, and the recognition of such companies by the ethnic minorities of this territory. The socioeconomic indicators of the area reveal that 37% of the population is of African descent and the economy is solid and versatile, thanks to a production structure based on diverse sectors, such as industry, tourism, commerce, and international sea trade logistics derived from its strategic location on the Caribbean Sea, to the north of South America and in the center of the American continent. Nonetheless, the city is one of contrasts and inequality in income distribution, with a poverty rate of 27.0% (http://www.cartagenacomovamos.org/nuevo/wp-content/uploads/2014/11/Presentacion-Calidad-de-Vida-2017-FINAL.pdf) (accessed on 7 October 2019).

The implementation of CSR in regions with a large percentage of Afro-descendant or indigenous ethnic minorities, as is the case of Cartagena de Indias, has to meet a special formal and documentary requirement, that of obtaining the so-called social license to operate, which refers to the acceptance of the community for the company's business to be conducted transparently and under a voluntary agreement; in other words, the arrangements are socially legitimate. Obtaining this license requires a process of interaction and communication with the communities in question, known as prior consultation. These processes and their supporting documents are studied so the parties can be fully informed of how these large business organizations carry out their CSR models, their contribution to the communities, and how these impact their performance as a result of being recognized by such communities.

The recognition and the community's perception of the firm as an agent for change will affect the success of the CSR actions [16,17] and this is related to society's awareness of how firms, as well as conducting their business activity, also bolster the development of the basic services provided by the public administration [18,19].

The aim of this article was to analyze the association between the CSR actions implemented by firms to improve the living conditions of indigenous communities located in Cartagena de Indias, as regards education, health, culture, employment, leadership, and other aspects related to economic, social and cultural development, and the communities' recognition of the firms. Law 21/1991 [20] on Prior Consultations in the Colombian Caribbean provides the framework under which companies can obtain legitimacy as agents involved in the economic, social, and cultural development of the territory, safeguarding the environmental protection of the areas in which they operate. Our analysis draws on Stakeholder Theory and Social Legitimacy Theory [21,22].

The article is structured as follows: This introduction is followed by a section describing the theoretical framework on which the research is based. The Section 3 details the

proposed hypotheses. The Section 4 describes the methodology used, which is followed by a Section 5, which are discussed in the Section 6. The final Section 7 presents the conclusions, the limitations of the work, and future research lines. The article finishes with a complete bibliography.

## 2. Corporate Social Responsibility and Corporate Legitimacy—Theoretical Framework

CSR has shifted the classic economic vision of a firm's responsibility to that of maximizing the profits of its shareholders [23] towards less limited, broader perspectives that include the wishes, aspirations, and demands of all the parties that might have a certain interest in, or be affected by, an organization's actions. This has obliged firms to include stakeholders' wishes in their goals and decisions [24,25]. This broader understanding of the interest groups to be satisfied has increased the costs of more responsible organizations [26].

Drawing on stakeholder theory [13,26–34], we analyze the role of companies as active agents of change and progress in the areas under study. Such firms cannot act in isolation but must create relationships with all the agents involved in the local environment, where their objectives and the way they operate must respect the social norms and the bastions of the community.

Stakeholder theory sets out the framework under which firms implement CSR initiatives. Interaction with the primary agents, those with whom they carry out transactions—clients, suppliers, and workers—allows them to negotiate compensations for the possible negative externalities generated by their production activity. Business organizations, however, also have an impact on secondary agents, such as local communities or society in general, which are difficult to compensate for if not through CSR actions. These actions generate a competitive advantage [35]. Budianto and Suyono [36] state that family businesses do not consider CSR to be a priority, but for large or multi-national companies, it has a significant influence [37].

A firm's CSR-related decisions and actions in any of the three dimensions—economic, social, and environmental—create an image in the market that shapes its reputation among local communities and potential employees and clients. A greater intensity of CSR investment has a positive effect on local communities' recognition of the firm and creates greater value for all the agents involved [38].

A company thus becomes an institution with a function that goes beyond the merely economic realm, and its behavior is affected by internal and external factors. The company's search for recognition and legitimation in the environment in which it operates emerges as a primary objective for survival. This brings us to stakeholder theory, which shows how the demands of the community should be attended to by firms, and also to legitimacy theory, which provides the foundations for our research hypotheses, focusing on a company's recognition and acceptance as a "valid" agent in the environment.

Legitimacy theory enables us to construct a model with which to evaluate the level of recognition achieved by these companies in the indigenous and Afro-descendant communities in the areas in which they operate. This recognition depends on these communities' satisfaction with the improvements in their living conditions derived from firms' economic activities [12,39–48].

Legitimacy is the condition or stature stemming from the coherence between a corporation's system of values and the society of which it forms part [49], with this coherence being what facilitates society's recognition. Shocker and Sheti [50] suggested that the survival or growth of such organizations depends on their aims being socially desirable and economic and social or political benefits being distributed to groups from which they derive their power. In addition, given that neither the sources of power nor the need for the services offered by organizations is permanent, these organizations should constantly overcome tests of legitimacy and relevance to show that society requires their services and that the groups profiting from the benefits of these institutions have the approval of society. When there emerges actual or potential disparity between what society expects of a firm and what it perceives, the organization's legitimacy, and thus its survival, comes under threat [51].

Faced with such a situation, companies are obliged to design strategies to repair their status [49,52–54].

Scherer and Palazzo [55] proposed a fundamental shift "from an output and power-oriented approach to an input related and discursive concept of legitimacy" where legitimacy is "socially and argumentatively constructed by means of considering reasons to justify certain actions, practices, or institutions and is thus present in discourses between the corporation and its relevant publics". Thus, a firm's acceptance depends on the process of communication and negotiation between the organization and the affected agents in society [56]. Olateju et al. [57] find legitimacy theory useful in explaining why companies engage in corporate social responsibility (CSR) activities and how they communicate these activities to stakeholders. However, there is a need for more research to better understand the causal mechanisms linking CSR to outcomes and to bridge the gap between theory and practice.

Law 21/1991 [20] on Prior Consultations in the Colombian Caribbean is intended to safeguard the process of communication with the local community and enable the company to understand their concerns and demands and negotiate with the community the commitments to be met. By including input from civil society, local community groups, and the public sector, firms are better able to understand the social and business environments [58]. This capability enables them to generate bottom-up development and build on the existing social infrastructure. If these groups, namely civil society, local community groups, and the public sector, perceive the company's legitimacy to have been breached and this breach is sufficiently extensive and continues over time, it will eventually lead to the disappearance of the business. They can revoke the license to continue operating in the community, which may lead, for example, to the loss of clients, state sanctions, impositions from suppliers, or a fall in the supply of work [59]. In the region under study in the present work, the situation could be even more serious, given that Law 21/1991 on prior consultations establishes that the failure to respect the agreement entails the license not being awarded and thus the business project cannot be initiated.

The following proposed hypotheses and their analysis will allow us to determine whether a firm's recognition as a valid agent in a territory depends on the results obtained and the impacts on the development of the CSR actions enshrined in the license. We will also establish the level of a firm's recognition and whether this depends on their level of participation and engagement with the processes of change.

## 3. Different Approaches to Corporate Social Responsibility for Business Legitimacy—Literature Review and Hypothesis Proposal

The three dimensions of CSR—social, economic, and environmental—provide a wide range of possible actions that companies can implement. Companies' CSR policies vary greatly in orientation and their content is partly explained by how developed the country in which they are located is and they evolve in line with changes in the development conditions in the region [13]. This may be due to differences in the needs and demands of the population of each country or region. The search for a fit between what communities desire and the CSR actions implemented guides the decisions made in this respect [60].

The sociodemographic and economic situation of the Colombian Caribbean creates a scenario of diverse needs in basic infrastructures, education, healthcare, and culture, which arguably stem from the State's inability to cover such fundamental needs and rights. The firms in the region are not responsible for these problems nor do they have the resources to prevent or solve them, but consolidated companies can have a great impact on the social environment, much greater than that of philanthropic institutions or organizations [61]. In response to the lack of coverage of basic needs, private companies, rather than concentrating on their roles as generators of wealth, undertake to solve, mitigate, or contribute to the development of communities in questions of health, education, and income by means of productive inclusion as employers and suppliers of vulnerable populations, through solidarity and other practices related to fundamental rights. By doing so, companies can

gain the acceptance and support of different stakeholders and enhance their reputation and legitimacy in society [62].

Any CSR action implemented by a company, be it internal or external, generates expectations in local communities, and the actions and the results obtained may improve the communities' evaluation of the companies, legitimizing, in turn, their existence. The proposal for a comprehensive CSR model in the area under study requires consideration of the impact of the three previously mentioned CSR dimensions on a company's recognition, which leads us to propose partial hypotheses for the different lines of CSR action.

The hypotheses to be verified are all expressed positively, considering that companies' actions are guided by the demands the local communities put forward in the prior consultations, which are explicitly set out in the social licenses. Thus, as the actions included in the license are tailored to the desires of the local communities, the legitimacy of the firms is expected to be enhanced. Many studies have described the CSR of firms in different geographical regions and sectors [63–65]. Their findings suggest that companies prioritize certain lines of action over others depending on their location, sector, and their system of values and preferences. Lack of planning leads organizations to respond to specific emerging demands without a clear CSR policy. The review of the existing literature, and in accordance with the provisions of Law 21/91 on Prior Consultation [20], establishes as the main policies on which companies' CSR should focus the improvement of health, education, economic conditions, professional skills, the strengthening of the supply chain, the improvement of environmental conditions, the development of culture, and the strengthening of community leadership. The development of these policies can involve a positive relationship between the impact of CSR and communities' recognition of firms in the Colombian Caribbean under study. We also expect to find that such recognition will be greater if the actions proposed match what the communities expect of the firms.

In the following subsections, we set out the review of the literature that will lead to the formulation of each of the hypotheses put forward in this research.

### 3.1. Corporate Social Responsibility Focused on Health Improvement and Its Influence on Company Legitimacy

One of the primary lines of actions firms include in their CSR programs is that related to health conditions, be it through investment in basic healthcare structures, provision of health services, campaigns to fight against malnutrition or to promote healthy behaviors and public health, or others, such as the training of healthcare professionals. Many of the previous studies focused on companies located in underdeveloped countries have described the CSR actions undertaken and the recognition that the health measures have brought to the firms. Many of these actions undertaken in health protection are related to investment in infrastructures, aimed at water purification, waste management, building of hospitals, healthcare or health insurance in the workplace, and ensuring the presence of health and safety certification [64].

The main recommendations for firms to achieve legitimacy, according to Gifford et al. [46] in a study on gold mining companies, is to help communities train and retain healthcare professionals, giving rise to training programs for health workers to ensure sufficient coverage and replacement for the provision of related services.

Ioannou and Serafeim [65], in an analysis of CSR in firms in Nigeria, found their CSR actions are clearly oriented towards enhancing healthcare and education, covering the population's basic needs, and thus compensating for the lack of action by the Nigerian political system.

In addition, Roy et al. [66] report that social sustainability can be achieved through investment in education, health, nutrition, and community participation. Dutta et al. [67], in their study on company CSR initiatives in India, concluded that health is the field on which most actions focus, followed by education.

Hiswàis et al. [68] argued that businesses have an opportunity to help address the most important societal challenges, especially the social determinants of health, which

are the root causes of inequities in health. By doing so, businesses can make a positive impact on society and contribute to the legitimacy of companies as members responsible for improving the health of the population.

In other cases, it is considered that CSR can contribute to the promotion of health in the workplace and the creation of healthy work environments, which can have a positive impact on the health of workers and, by extension, the health of the general population [69,70]. Within this framework, we put forward the first hypothesis:

**Hypothesis 1.** *Including actions to improve health in CSR initiatives has a positive impact on the recognition of companies by communities in the Colombian Caribbean.*

### 3.2. Corporate Social Responsibility Focused on Education Improvement and Its Influence on Company Legitimacy

As mentioned, many works in the CSR literature have referenced the impact of investment in education on the development of countries firms are located in. Different socio-political scenarios and sectors all show a positive relationship between actions in this dimension (with some of the specific actions analyzed in each study being different) and firm legitimacy.

Goodland [71,72] argues that education is a key factor in promoting environmental sustainability. According to him, building human capital through education and training is one of the main means to accelerate the transition to population stability and renewable energy. Furthermore, he stresses the importance of investing in human resource development, with special attention to girls' education, as the single most important measure for both development and the promotion of sound long-term environmental policies. In short, education is fundamental to fostering environmental sustainability and must be a priority in efforts to achieve it.

Griesse [73] mentions that Brazil has high standards of education for only a small portion of the population, while a significant number still live in precarious conditions. He also points out that, although almost 97% of children aged 7–14 attend school, problems related to the quality of education persist, such as absenteeism, low achievement scores, illiteracy, and ill-prepared teachers. It can therefore be inferred that education is an important factor in the struggle for social and economic equality in Brazil [74].

Newenham-Kahindi [75] describes the case of the Barrick company in local communities in the Lake Victoria area in Tanzania, which developed a new policy on education support using royalties.

Mena et al. [76] mention that after receiving education in basic reading, writing, and mathematics, the women of Arzu are no longer so easily exploited by unscrupulous carpet buyers. In addition, alternative learning centers are established to bridge the gap between the community and the government in places where publicly funded schools are lacking.

Arato [77] mentions that one of the most popular CSR strategies implemented by companies is to provide training and educational support to members of the community.

Within this framework, we put forward the second hypothesis:

**Hypothesis 2.** *Including actions to improve education in CSR initiatives has a positive impact on the recognition of companies by communities in the Colombian Caribbean.*

### 3.3. Corporate Social Responsibility Focused on Economic Conditions and Its Influence on Company Legitimacy

Improving the economic conditions of the population in the territories where firms are located is an element commonly demanded by communities, especially when these CSR actions are undertaken in countries with high poverty rates and when the groups affected are ethnic minorities at risk of exclusion. Economic improvement emerges when firms generate employment and train the population in employment skills and entrepreneurship.

Newenham-Kahindi [75], in a study on a sample of multinational enterprises, reports they play a significant role in achieving social sustainability, working in cooperation with

multiple agents, local communities, and local governments and providing solutions for problems in their areas, such as high unemployment, social conflict and crime, tropical disease, gender inequality, youth training, and the promotion of local entrepreneurial activity.

Lee and Yoon [78] argue that CSR can include initiatives that seek to improve the well-being of the community in which the company operates, which could have implications for the income level of the population. For example, a company could implement training and employment programs for disadvantaged people, which could improve their ability to earn an income.

Other authors suggest that companies should be held accountable for their tax practices and that governments should take steps to ensure that companies pay their fair share of taxes. This would help to address income inequality by ensuring that everyone contributes to the public good and that governments have the resources they need to provide public services and infrastructure [79].

Adomako and Tran [15] argue that entrepreneurs should be encouraged to pay attention to achieving CSR legitimacy and integrating into the local community to foster responsible entrepreneurship. The empowerment of entrepreneurship is understood to improve the economic conditions of the population.

The following hypothesis refers to the relationship between economic conditions and the recognition of firm legitimacy in the community.

**Hypothesis 3.** *Including actions to increase economic income in CSR initiatives has a positive impact on the recognition of companies by communities in the Colombian Caribbean.*

*3.4. Corporate Social Responsibility Focused on Skills for Employability and Its Influence on Company Legitimacy*

Training for employment can increase opportunities in the community and improve their economic conditions.

Some authors underline the prioritization of internal CSR actions, which mainly affect employees, over external and environmental activities. This is the case of the work by Vives et al. [80], which in an analysis of the perspectives of investment in internal, external, and environmental actions by SMEs in the Caribbean over a period of 3 years, finds that CSR was focused on improving employees' health and wellbeing, training and participation in the business, the generation of opportunities, and the establishment of relationships with employees' families. The formation of human capital trained in CSR is one of the activities offered in the "Consolidation and Scalable Projects" program of FUNDEMAS in El Salvador.

Ioannou and Serafeim [65] find that training workers in skills is one of the most important variables for the measurement of the impact of CSR in the society where a firm operates.

Gallardo et al. [81] consider that companies that work on building human and relational capital can generate competitive advantages, which can lead to greater legitimacy compared to the competition. In this sense, it could be argued that companies that implement CSR and IC practices can improve their ability to attract and retain talent, which could contribute to the improvement of employability skills.

In addition, Nejati and Shafaei [82] suggest that CSR initiatives can improve talent attraction and retention, which can lead to a more qualified and engaged workforce. In addition, CSR initiatives can enhance the company's reputation, which can attract potential employees looking to work for socially responsible companies. However, it is important to keep in mind that the authenticity of CSR initiatives is crucial for employees to perceive them as meaningful and for them to have a positive impact on employee retention and engagement.

In their study on rural areas in Latin America, Arato et al. [77] suggest that the second most popular CSR strategy is the training of community members in the skills needed for technical jobs.

Within this framework, we put forward the fourth hypothesis:

**Hypothesis 4.** *Including actions to improve training for employment in CSR initiatives has a positive impact on the recognition of companies by communities in the Colombian Caribbean.*

*3.5. Corporate Social Responsibility Focused on Improving Local Supply and Its Influence on Company Legitimacy*

Creating a solid relationship with the company's local suppliers leads to a firm's enhanced competitiveness through competitive advantages in the supply of materials and products, avoiding foreign competition.

Cuervo-Cazurra and Dau [83] mention that the study focuses on the impact of pro-market reforms on the profitability of companies in developing countries. It is possible that the implementation of CSR practices in the supply chain could be a way to improve the profitability of companies in the long run, but further research would be needed to assess the relationship between CSR and supply chain improvement.

Urdaneta [64] argues that competitive pressure forces firms to increase efficiency and seek subcontractors for the provision of complementary inputs. Thus, suppliers become an instrument for local development, contributing to changes in attitudes and beliefs in the community, improving efficiency in the provision of products and services, and driving development and increasing the wellbeing of the poorest groups in developing countries [84].

They are thus seen as key actors in the promotion of sustainable development, especially when government initiatives are insufficient [85]. Other works, however, such as that by Sámano et al. [86], argue the opposite.

Within this framework, we put forward the fifth hypothesis:

**Hypothesis 5.** *Including actions to develop local suppliers in CSR initiatives has a positive impact on the recognition of companies by communities in the Colombian Caribbean.*

*3.6. Corporate Social Responsibility Focused on Improving Environmental Conditions and Its Influence on Company Legitimacy*

As discussed, actions aimed at protecting and enhancing the environment represent one of the priority areas of CSR. This focus on environmental sustainability is found in both developing and developed regions [87].

Although these voluntary initiatives implemented to attend to the needs of the population are costly, a growing number of firms across many sectors and parts of the world have seen how their socially responsible practices, such as pollution prevention, energy efficiency, design of environmentally friendly products, supply chain management, and actions for sustainable agriculture, have had a positive impact on profits [88].

These measures are sometimes implemented because of demands by the administration, to meet legal requirements, or are driven by incentives. Urdaneta [64] mentions that social responsibility implies incorporating environmental considerations in decision making and being accountable for the impacts of decisions and activities on the environment. Therefore, it can be inferred that some concrete actions to safeguard the environment could include the reduction in greenhouse gas emissions, proper waste management, and biodiversity conservation.

Griesse [73] is of the same opinion and argues that these actions are undertaken as part of company policy, either because they wish to minimize environmental impacts forms part of their management guidelines or because they recognize that sustainable behavior improves performance by generating powerful competitive differentiation and improves their efficiency and reputation.

Joyner and Payne [89] mention the importance of ethics, values, and corporate social responsibility including the consideration and preservation of the environment as one of the key areas of focus.

Firmansyah and Estutik [90] suggest that companies with high environmental responsibility performance and that comply with applicable regulations can establish stronger connections with the public and improve their ethics. In contrast, non-sustainable be-

haviors are rejected by local communities, and thus, our sixth hypothesis refers to the following relationship:

**Hypothesis 6.** *Including actions to improve the environment in CSR initiatives has a positive impact on the recognition of companies by communities in the Colombian Caribbean.*

*3.7. Corporate Social Responsibility Focused on Improving Culture and Its Influence on Company Legitimacy*

Culture is a further focus of CSR action. McNamara et al. [91], in their study on CSR across different countries, define the need for a culture and value fit between firms and communities. As in organizations, the so-called person-culture fit mentioned by O'Reilly et al. [92] helps employees identify with their company and thus communities can be expected to identify more with firms that attempt to fit their values to those of the minorities in the territory.

Urdaneta [64] mentions that one of the objectives of social responsibility is to universalize culture and sports, and proposes some concrete actions to achieve this, such as promoting cultural development in the organization, supporting communities in cultural activities, developing cultural spaces in the communities, and promoting within the suppliers the support for works to improve sports facilities in the communities.

The abovementioned studies suggest that the cultural fit between the local community and firms can enhance ethnic minorities' perception of the business that operates in their area. Thus, our seventh hypothesis is:

**Hypothesis 7.** *Including actions to Afro culture in CSR initiatives has a positive impact on the recognition of companies by communities in the Colombian Caribbean.*

*3.8. Corporate Social Responsibility Focused on Improving Leadership Community and Its Influence on Company Legitimacy*

There is a large body of literature on the role of corporate leaders in the process of communication with firms in promoting CSR. Some authors hold that the key to this strengthening lies in dialogue with stakeholders through creating channels of communication and developing alliances [77].

Martin [93] suggests that each CSR action requires a corporate coalition and an intrinsically motivated community leader, with energy, vision, and the communication skills needed to convince company chiefs. The works by Yu [94], Kemp et al. [34], and López et al. [95] also defend the same premise.

Syna et al.'s [96] emphasis on the importance of promoting gender equality, diversity, and inclusion in local politics can be seen as relevant to the broader goal of creating more inclusive and equitable societies, which could be supported by companies as responsible members of the community.

Lanza [97] suggests that there is a relationship between the economic outlook of a state, its security outlook, cultural issues, and how these relate to ethnic minority oppression. Watkins et al. [98] conclude that family CEOs favor social performance, while non-family CEOs decrease it. Therefore, the paper highlights the importance of family CEOs in promoting CSR practices and contributing to the leadership of citizens in terms of social and environmental responsibility.

Within this framework, we put forward the eighth hypothesis:

**Hypothesis 8.** *Including actions to strengthen community leadership in CSR initiatives has a positive impact on the recognition of companies by communities in the Colombian Caribbean.*

## 4. Methodology

First, we used a database from the Colombian Ministry of the Interior to identify projects enacted by companies that signed prior consultation agreements with ethnic minority communities in the Colombian Caribbean between 2011 and 2016, using a research proto-

col to determine the significance of the project for the region under the 2008–2032 Cartagena and Bolívar Regional Competitiveness Plan. We selected four firms as particularly representative due to their engagement in communities, their investment capital, the territories affected, and their biodiversity. These cases were Serena del Mar from the urbanization and construction industry, Karibana Beach Resort from the recreational tourism sector, Puerto Bahía from the port logistics sector, and Surtigas from the energy and public services sector.

Serena del Mar is being built on the north coast of Cartagena, a new city designed by world leaders in urban planning and landscaping, which is based on the magical enclave of the Colombian Caribbean landscape. Serena del Mar is a bold proposal that takes a new look at the urban planning needs of the region, preserving and highlighting the natural diversity that enriches its surroundings. The new city incorporates global innovation in balance with the essence of deep-rooted culture and breathtaking geographic beauty, giving the region new opportunities in housing, health, education, entertainment, hospitality, and business. It is designed with citizens in mind, with a vision that maintains a balance between progress and roots, between tradition and the avant-garde. In the residential aspect, it will have a total of 17,000 homes for a population between 50,000 and 60,000 people. In recreation and infrastructure, it will have public areas for culture, entertainment and recreation, social clubs, marinas, canals, beaches, bike paths, and trails with more than 100 kilometers. Institutional infrastructure will include a 70,000 m$^2$ hospital, schools, and a transportation terminal for the city, to be built in stages over the next few years.

Puerto Bahía is a multipurpose port complex located on a 155-hectare lot on Barú Island in the Bay of Cartagena. The port complex will take advantage of the robust prospect of crude oil production in Colombia and the need for more and better facilities for its transportation and export. The value proposition presented by Puerto Bahía is not limited to its design. The port has a strategic location in the Bay of Cartagena, itself one of the largest trade centers in Latin America. Puerto Bahia's competitive advantages are evident in its marketing strategy. The port has an initial storage capacity for liquids of 3.3 million barrels and a general cargo handling facility for trans-shipment, dry cargo, and general bulk cargo. The port is optimally designed to meet current demand but also has sufficient space available for future expansion.

Karibana is a one-of-a-kind development with a wide range of entertainment and leisure activities. This project reflects the brand's intention to be present in each of the world's most important gateway cities and most sought-after leisure destinations. The project is located north of the city of Cartagena, between the towns of Manzanillo and Punta Canoa. In addition to representing a complete life experience, surrounded by the best facilities for the rest and peace of families in a natural and culturally rich environment, since 2008 it has been the project with the highest value in Cartagena. The investment for this project will be 150 million pesos, which is expected to be inaugurated in December 2016. Surtigas is a company with 43 years of experience, during which it has provided access to natural gas and associated services in Colombia to more than 550,000 households. The company currently operates in more than 120 towns in the departments of Bolívar, Sucre, Córdoba, Antioquia, and Magdalena, making it the natural gas distributor and marketer with the largest geographic area served in the national territory: 90,000 km$^2$. In these four decades, Surtigas has achieved 90% coverage in its area of influence, bringing energy solutions and well-being to thousands of families, 80% of whom belong to strata 1 and 2. It also provides solutions to 5531 commercial customers and more than 315 industrial customers.

With a CSR management system focused on sustainability, Surtigas aims to promote more equitable, livable, and viable scenarios in the country through a balance between the social, economic, and environmental aspects of its business actions [99].

A previous study analyzed the social license documentation from firms' prior consultation processes with communities affected by projects. Four cases were chosen based on the following criteria: Being part of a regional productive commitment; location along the metropolitan area of the city (locality), evidence of the magnitude of economic, environmental, and social impacts (communities and ecosystems); differential in Afro-descendant

communities (different communities impacted); and varying levels of progress in processes of Prior Consultation agreements.

Figure 1 shows a list of the communities affected by the activities of the companies under analysis. It can be seen that operations by Serena del Mar affected the communities Tierra Baja and Manzanillo del Mar; the activities of Karibana Beach Resort had an impact on the community of Punta Canoa; Puerto Bahía had a notable effect on the communities of Ararca, Bocachica, and Barú; and Surtigas carried out activities affecting Tierra Bomba and Caño de Oro, populations amounting to approximately 28,422 inhabitants as of the 2015 census. These communities are made up of Afro-descendant and Raizal ethnic minorities, which are the most representative in the Region. This has served to guarantee the participation of these ethnic minorities in the protection of their ethnic and cultural integrity before initiating projects to search for and exploit natural resources in their territories.

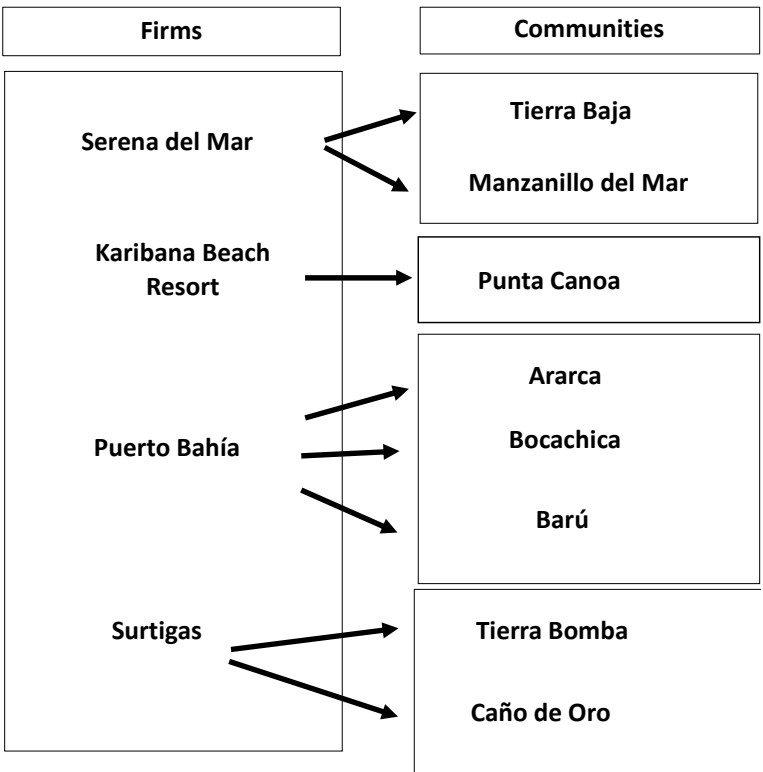

**Figure 1.** Firms and communities affected. Source: Own preparation.

As this work presents its results from the perspective of the communities, after determining the cases to be studied and identifying the groups or communities affected by the CSR actions, we designed a questionnaire to collect the information required to understand the communities' perception of territorial development and the impacts of the CSR activities agreed in the social licenses. To configure the sample, a non-probabilistic convenience sampling method was applied [100], where the inclusion criterion was the evidence of the representativeness of the respondent in the group of actors involved [101–103]. By representativeness we mean the capacity of certain people to embody the feelings and thoughts of the social segment to which they belong, so the researcher should select the person who offers an important opportunity for learning and providing information [104].

In this case, the interview was conducted with each main CSR manager with the prioritized ethnic minority communities, with good communication and fluency of content, achieving the joint construction of the meanings of the themes. A short, semi-structured questionnaire was used to collect all the impressions and perceptions of the manager, adding all the supporting documents provided by the company. The questionnaire was

thus the primary data source and was aimed at the members of the local community involved in the business projects selected. This questionnaire was completed by all the CSR managers of the eight ethnic communities. Their responses resulted in a sample of 98 interviews distributed as shown in Table 1.

**Table 1.** Distribution of sample by communities.

|  | Serena del Mar | Karibana Beach | Puerto Bahía | Surtigas |
|---|---|---|---|---|
| Tierra Baja | 11 | | | |
| Manzanillo del Mar | 16 | | | |
| Punta Canoa | | 5 | | |
| Ararca | | | 9 | |
| Bocachica | | | 13 | |
| Barú | | | 14 | |
| Tierra Bomba | | | | 22 |
| Caño de Oro | | | | 8 |
| **Total** | **27** | **5** | **36** | **30** |

Source: Own preparation.

The instrument used to collect the information and the dimensions it covered was based on a territorial development approach. We included dimensions related to basic needs, improvement of economic conditions, environmental impact, preservation of culture, and strengthening of community leadership.

Having collected the information, the data were evaluated under a legitimacy-recognition model.

A descriptive analysis of the results obtained in each community was conducted, focusing on the degree of recognition or legitimacy the communities attributed to the firms in relation to their CSR actions, depending on the independent variables included in the research.

We also conducted a binomial logistic regression, a method that measures the contribution of different events to a simple event outcome. This enabled us to assess the level of recognition of the agent, namely the firm, as a result of the effects of the CSR practices in the communities depending on the variables involved in their vision of development.

The choice between binary logistic regression has been determined by the nature of the variables involved and the type of question you are trying to answer. Table 2 lists the dependent variable and the independent variables used in the analysis as dichotomous and takes the value of 0 when the variable is present and 1 if not. This process is known as binomial as it only has two possible outcomes, with the probability of each one remaining constant in a series of repetitions.

**Table 2.** Variables and statistical methods applied.

| Dependent Variable | Independent Variables | Statistical Method |
|---|---|---|
| **Firm's recognition (FR)** | Contribution to health (H)<br>Improvement in education (E)<br>Income (I)<br>Training for employment (T)<br>Development of suppliers (S)<br>Environmental wealth (EN)<br>Appreciation of Afro descendant culture (C)<br>Strengthening of community leadership (L) | Binomial logistic regression |

Source: Own preparation.

In our case, the dependent variable was the firm's recognition (validity of the agent), being 1 if the firm was recognized and 1 otherwise. The independent variables were the evaluations of the effects of the CSR actions enacted by the firms: Economic, environmental, and sociocultural dimensions, which we expected to represent the factors explaining the

firm's recognition. The equation below represents the model used, which represents the dependent variable as a function of all the independent variables used and coded as shown in Table 2.

$$FR = \beta_0 + \beta_1 H_i + \beta_2 E_i + \beta_3 I_i + \beta_4 T_i + \beta_5 S_i + \beta_6 EN_i + \beta_7 C_i + \beta_8 L_i + \varepsilon_i$$

## 5. Results

The results of our descriptive analysis for each of the selected firms are detailed below.

The Serena del Mar Company mainly operates in the communities of Tierra Vieja and Manzanillo del Mar. All the respondents from the community of Tierra Vieja consider Serena del Mar to be a recognized member of their society. Only 27.3% consider the company to have contributed to improvements in health, while all the respondents feel that improvements have been enacted in education. In this sense, 36.4% consider that improvements have been made in education infrastructures, and the same percentage think educational processes have been enhanced. However, more than 90% of respondents report no changes in their level of income and that the impacts on the environment are negative. A total of 73% of the respondents consider the firm has prioritized the respect for Afro-descendant and Raizal culture in conducting their activities. More than two-thirds of the respondents agree that the company has shown its commitment to developing entrepreneurship in the community, above all by promoting training and supporting initiatives through seed capital. Furthermore, almost all of the citizens interviewed report that the firm has contributed to strengthening community leadership capacities and co-managing questions of interest for the community with its leaders. Most of the respondents, however, made no comment on the firm's support for the development of local suppliers.

Of the respondents in the community of Manzanillo del Mar, 73% consider Serena del Mar to have helped manage and support key aspects related to basic health needs, while almost all the participants think that the firm has generated improvements in schools with regard, mainly, to educational processes. A total of 40% report their level of income has improved thanks to the company's actions, while 20% say their income has fallen due to the loss of savings. Almost 60% of the population consider the company's impact on the environment to be fully positive, and practically all the respondents think the Afro-descendant and Raizal culture has been respected. More than half feel the firm has contributed to entrepreneurship in the community, especially by giving local suppliers the opportunity to provide goods to the company. In addition, 30% think the firm has helped strengthen community leadership capacities by means of economically supporting the actions of the Community Council and the Community Action Board and co-managing questions of interest with local leaders. Consequently, 86% of the respondents consider that Serena del Mar has become a recognized member of their community.

In the case of Karibana Beach, the Punta Canoa community considers (60%) that the firm has contributed to managing and supporting aspects of healthcare that have benefited the citizens. Their opinion is unclear as regards the improvements implemented by the company in schools and educational institutions, as is their opinion about the company's impact on the environment (20% very positive vs. 20% negative). A total of 40% think their income has declined due to the loss of savings, and the same percentage considers the firm to have respected the Afro-descendant culture and contributed to developing entrepreneurship in the community, primarily as regards training. Of the respondents, 40% say the company has helped strengthen community leadership capacities, while 20% feel they have had no impact in this respect. There is no clear opinion, either, about the firm's support for the development of local suppliers. Overall, the community feels that Karibana Beach is not a recognized member of their society.

In the case of Puerto Bahía, one-third of the respondents in the community of Ararca think the firm has contributed to managing and supporting health-related elements, while almost all the respondents believe it has improved the schools and educational institutions, with particular reference to processes and infrastructures. The firm's actions have not

led to increased income and half of those interviewed think their income has remained the same. A total of 77% report negative impacts on the environment, while more than 50% think that Afro-descendant customs and culture have been respected. The same percentage of respondents have no opinion about the firm's commitment to developing entrepreneurship in the community, while the other 50% believe that Puerto Bahía has contributed to this aspect by promoting initiatives to generate businesses and co-operatives. Almost half of those interviewed think the firm has had no impact on strengthening community leadership capacities, while two-thirds feel the company has boosted the development of local suppliers, stimulating and supporting local socio-environmental projects. Finally, 90% of the respondents report that Puerto Bahía is a recognized member of the community.

Almost two-thirds of the respondents in the community of Boca Chica consider that healthcare has been neither improved nor supported by Puerto Bahía, although 53% think the firm has helped improve education, especially with regard to educational processes and infrastructures. For 60%, income levels have been preserved and 40% consider the firm has had no impact on the environment. Almost 50% of those interviewed think respect for Afro-descendant culture has not been prioritized, while the same percentage had no clear opinion about the company's commitment to developing entrepreneurship in the community. Additionally, 40% consider the firm has had no impact on strengthening community leadership, while 46% believe the company has supported local suppliers, especially regarding the implementation of good practices and stimulating and supporting local socio-environmental projects. Overall, 40% consider the firm to be a recognized member of the community, 20% feel it is not, and 26% have no clear opinion.

In Barú, opinions are divided about CSR actions related to improving health in the community (28.6% consider there has been little or no contribution and the same percentage believe the company had made some contribution or a notable contribution in this sense). However, in the field of education, 78.6% consider the firm has helped improve the schools and educational institutions, especially regarding educational processes and the provision of technology. Almost two-thirds of the respondents believe income levels have remained unchanged, while 75% think the firms have either had no impact on the environment or the impact has been positive. Opinions are also divided about whether the firm has prioritized Afro-descendant culture in its activities.

In addition, 57.1% feel the company is committed to developing entrepreneurship in the community, with reference to bolstering skills training and the generation of opportunities for local suppliers, implementing good practices, establishing permanent links, and prioritizing purchases from local companies. Half of those interviewed feel the company has helped strengthen leadership capacities and half consider they have provided notable financial support for the actions of the Community Council and the Community Action Board. Finally, 85.7% of the respondents consider Puerto Bahía to be a recognized member of their community.

Finally, in the case of Surtigas, 82% of the respondents consider the company to have made no contribution to healthcare actions in the community of Tierra Bomba, with the same percentage having a similar opinion about the firm's impact on the schools and educational institutions in the territory. Half of the citizens interviewed think their income level has remained unchanged, while the other half have no clear opinion in this respect. A total of 59% have no opinion about the firm's environmental impact, while the other 41% feel the impacts have been negative. Of those interviewed, 68% think the Afro-descendant and Raizal cultures have been respected. The respondents have no clear opinion about the firm's commitment to developing entrepreneurship in the community, their support for local suppliers, or their contribution to strengthening community leadership capacities. Consequently, only 27% of the respondents consider Surtigas to be a recognized member of the community of Tierra Bomba, while 55% fail to recognize its legitimacy.

The respondents in the community of Caño de Oro consider Surtigas to have contributed nothing or very little to improving healthcare in the area (62.5%), while 75% think

that improvements have been made as regards the schools and educational institutions, especially as regards teaching materials and infrastructures. A total of 62% believe their income has remained unchanged or has declined, while 37.5% believe the company has had no impact on the environment and 25% believe the impact has been positive. In addition, 62.5% think the firms have prioritized respect for the Afro-descendant culture. A total of 50% of the respondents consider the company has shown commitment to fostering entrepreneurship, particularly as regards giving opportunities to local suppliers and stimulating and supporting local socio-environmental projects. Finally, the citizens interviewed feel Surtigas has had little effect on strengthening community leadership capacities. Overall, most of the respondents (75%) think Surtigas is an important member of their community.

The logistic regression statistical analysis revealed an association between the impacts of the different CSR dimensions and the firms' recognition among the community. The starting point was the following null hypothesis: The probability of a company being recognized cannot be predicted by the impacts of its CSR. The contrasting alternative hypothesis was that the probability of the company being recognized can be predicted by the impacts of its CSR.

The classification of observations (Table 3) shows the logistic regression model was able to correctly classify 93.1% of the cases in which the firm is a recognized member of the community and 48% of those when it is not recognized as such. Thus, the logistic regression analysis indicates the probability of the dependent variable performing correctly is 81.4%. In other words, when the results of the independent variables for the impacts of CSR are known, the probability of correctly predicting whether the company is recognized by the community is 81.4%.

**Table 3.** Classification of observations.

|  | Observed | | The Company Is a Recognized Member of the Community | | |
| --- | --- | --- | --- | --- | --- |
|  |  |  | Yes | No | Percentage Correct |
| Step 1 | The company is a recognized member of the community | Yes | 67 | 5 | 93.1 |
|  |  | No | 13 | 12 | 48.0 |
|  | Overall percentage | | | | 81.4 |

Source: Own preparation.

Table 4 shows the results of the logistic regression and the statistical indicators. Considering the value of Nagelkerke's R square, we can state that the independent variables explain 45.7% of the variance of the dependent variable. That is, the impacts of the CSR actions implemented in the territory explain the communities' recognition of the company.

**Table 4.** Logistic regression model.

|  | $\beta_i$ | Standard Error | Wald | Gl | Sig. | Exp(B) |
| --- | --- | --- | --- | --- | --- | --- |
| Improvements in health | 0.195 | 0.140 | 1.934 | 1 | 0.164 | 1.215 |
| Contribution to education | 0.480 | 0.169 | 8.055 | 1 | 0.005 | 1.615 *** |
| Increased income | 0.195 | 0.117 | 2.752 | 1 | 0.097 | 1.215 * |
| Training for employment | 0.486 | 0.188 | 6.687 | 1 | 0.010 | 1.625 *** |
| Development of suppliers | 0.077 | 0.105 | 0.527 | 1 | 0.468 | 1.080 |
| Environmental wealth | 0.261 | 0.121 | 4.606 | 1 | 0.032 | 1.298 ** |
| Afro descendant culture | −0.519 | 0.208 | 6.253 | 1 | 0.012 | 0.595 ** |
| Community leadership | −0.092 | 0.124 | 0.546 | 1 | 0.460 | 0.912 |
| Constant | −4.578 | 1.458 | 9.863 | 1 | 0.002 | 0.010 *** |
| Chi-square | | | 36.142 ** | | | |
| Log likelihood | | | 74.568 | | | |
| Cox and Snell R square | | | 0.311 | | | |
| Nagelkerke's R square | | | 0.451 | | | |

*** sig. 99%; ** sig. 95%; * sig. 90%. Source: Own preparation.

Table 4 shows the independent variables that improve the probability of the dependent variable predicting correctly. The variables that are significant at a confidence level of 99% to 90% have a substantive effect on recognition.

The results for hypotheses related to basic needs—health and education—are dissimilar. While improvement in health was not a significant predictor of recognition, leading us to reject $H_1$, the contribution to education was significantly associated with the dependent explanatory variable, which means $H_2$ is accepted.

Regarding the second block of hypotheses related to economic conditions and their association with recognition, the logistic regression model shows that increased income and the creation of opportunities through training for employment and entrepreneurship have a significant effect on a firm's recognition. The results, however, do not evidence such a relationship between the development of suppliers and the recognition of the company in the community. Thus, $H_3$ and $H_4$ can be accepted, while $H_5$ is rejected.

The logistic model shows a significant association between improvements in the environment and the dependent variable, thus confirming $H_6$.

Finally, the testing of the two hypotheses on the integration of business projects in the communities through strengthening the Afro-descendant culture and community leadership yielded contrasting results. While actions related to enriching the local culture generate recognition for the company, leading us to accept $H_7$, strengthening community leadership shows no significant association with a firm's recognition, meaning $H_8$ is rejected.

Table 5 details the hypotheses accepted and rejected.

**Table 5.** Acceptance and rejection of the proposed hypotheses.

| | |
|---|---|
| H1: Improved Health Impacts on Recognition | Rejected |
| H2: Improved education impacts on recognition | Accepted |
| H3: Increased economic income impacts on recognition | Accepted |
| H4: Improved training for employment impacts on recognition | Accepted |
| H5: Improvements for suppliers' impact on recognition | Rejected |
| H6: Improved environmental conditions impact on recognition | Accepted |
| H7: Strengthening the Afro descendant culture on recognition | Accepted |
| H8: Strengthening community leadership impacts on recognition | Rejected |

Source: Own preparation.

The results of the model show that the most robust associations with recognition are those of training for employment (1.625) and improved education (1615). These findings are consistent with the evidence found in the social licenses of the cases under study. The communities prioritized these actions in the processes of prior consultation with the companies so they would be included in the social licenses.

## 6. Discussion

The results for our first hypothesis suggest there is no positive relationship between companies' health-related initiatives and their recognition in the local communities. The only activities recognized by the community in this respect are those enacted in Manazanillo del Mar by the Serena del Mar Company. This runs counter to the findings of other studies, such as that by Gifford et al. [46] on the gold mining industry in Africa, in which one of the main recommendations to achieve firm legitimacy is to help communities train and retain healthcare professionals. Thus, there is an established positive relationship between initiatives targeting improvements in healthcare and their impact on the recognition of companies. The work by Hiswàis et al. [68] presents similar evidence.

In her work on CSR, Urdaneta [64] finds that the health-related actions most widely demanded of companies in the Venezuelan oil industry are those associated with safeguarding the health of employees and their families and those aimed at supporting public health works that enhance the living conditions of the local communities. Other authors have related the positive image of firms with projects intended to fight against specific diseases [69,70]. Such is the case of the studies that present initiatives to combat HIV [67].

In our view, as safeguarding health is a basic need, we understand that the communities included in our study consider the initiatives focused on health-related improvements do not serve to improve the image or enhance the recognition of companies given that health care is indeed a basic right. While the failure to safeguard health in the community is related to a negative image of organizations, their image is not improved, however, when they do guarantee this fundamental right of communities.

As regards our second hypothesis, our results confirm a positive relationship between greater recognition of the companies and improvements in education in the communities in which they operate. It is worth noting that this variable has the greatest impact on the dependent variable of all the independent ones used in the model. This is consistent with other works conducted in rural areas of Latin America, which analyzes CSR actions thus helping to eradicate illiteracy in the population. Arato et al. [77] underline a strategy that has also been reported but is implemented by a smaller number of companies due to the size of the investment required, which is to provide infrastructure, materials, or even the full construction of school facilities for their stakeholders. In rural communities, Griesse [73], in a study on CSR in Brazil, describes how education in public schools continues to be precarious despite attempts by the government to improve standards. The Bolsa-Escola (School Scholarship) program implemented by Cardoso in 1995 provided a small monthly payment for low-income families whose children were attending school. This idea succeeded in keeping children in school and was selected by the United Nations as a model to be implemented in other countries. While almost 97% of children aged 7–14 reportedly attend school, there exist problems related to the quality of education, illiteracy, and poorly prepared teachers [74].

Similar educational projects have been undertaken in rural areas of Africa. Newenham-Kahindi [75] describes the objective as increasing the level of education in the impact area, funding the construction of classrooms and housing for teachers, and providing school stationery and building materials for various schools in the area. In rural areas of India, studies have also reported examples of companies empowering the population by promoting education [76].

The confirmation of our third and fourth hypotheses evidences the positive relationship between improving the image or recognition of companies and the implementation of activities focused on increasing economic income and training for employment in local communities [78]. These findings are consistent with the study by Newenham-Kahindi [75] as one of the companies' primary goals in CSR is to attempt to improve business and financial management skills among local interested individuals, with the aim of improving their income and living standards and attracting talent [82]. One way to achieve this is by developing entrepreneurial projects in rural communities, mainly focused on the primary sector [15]. As regards improving training for employment, our results coincide with those of the studies by Ioannou and Serafeim [65], which underline that one of the most popular strategies, as reported by the companies analyzed, was to provide training and support education and training for community members. The most popular form of support in terms of education was the provision of work and technical training, as well as workplace safety training for employees [77].

Regarding the link between a company's recognition and improving relationships with local suppliers, the responses to our survey do not suggest there exists a significant association, as confirmed in our analyses for the fifth hypothesis. The only exceptions are in the community of Barú in the case of the Serena del Mar Company, and in Caño de Oro in the case of Surtigas. These findings are consistent with those of Sámano et al. [86] in their work on environmental responsibility, which report that suppliers and companies present a good relationship, but this is focused only on the buying and selling of their products, without considering this brings companies greater recognition in the community. In contrast to our findings are those of Urdaneta [64], who, in their studies on CSR in the Venezuelan oil industry, find that firms in the sector enact policies that encourage suppliers

to participate in social development in local communities through programs to promote values such as solidarity, co-operation, reciprocity, and equality.

Actions intended to improve the environment generate a positive impact on a firm's recognition, as demonstrated by our sixth hypothesis. This finding is consistent with those obtained in other studies, such as those by Griesse [73], Holtbrügge et al. [87], Urdaneta [64] or Firmansyah and Estutik [90], where it is reported that firms have pursued environmentally friendly practices by focusing on more efficient uses of resources and energy and by recycling. Most of these efforts have been designed to reduce waste, reduce risks, and improve efficiency within the firm. External pressures appear to be the most effective method to force companies to implement CSR practices that benefit the environment. This also implies that communities and governments should implement strict regulations and other political incentives to encourage CSR practices, which, in turn, would avoid the undertaking or not of these actions endangering firms' reputations in local communities.

The analysis of our seventh hypothesis on the cultural fit between the local community and the company has a positive impact on ethnic minorities' perception of the firms that operate in their territory. This is in line with the findings of Urdaneta [64] on the positive effects on the recognition of firms of cultural activities related to local communities' traditions. This is also reported in the works by O'Reilly et al. [92] and McNamara et al. [91].

Finally, no positive association was found between implementing initiatives to strengthen leadership in communities and improved firm legitimacy, as demonstrated by the findings for our eighth hypothesis. This relationship was only revealed among the community of Tierra Baja in the case of CSR enactment by Serena del Mar. These findings contradict those of other research reporting that strengthening community leadership enhances the recognition of firms that operate in such areas [77,93,98]. Kemp et al. [34], in their study on the mining industry, highlight the need to encourage leadership, empowering the community and including firms' internal audit processes. Similarly, Lopéz et al. [95], in their study on CSR in fumigation companies, describe how developing and engaging with the community impacts positively on the perceived level of firms' CSR enactment. These authors suggest that developing community leadership might have an effect on firm legitimacy through communities feeling they are valued and empowered in their relationships with the companies. Yu [94] describes the example of how the footwear manufacturer Reebook encourages empowerment in the community through CSR actions related to employee training, educating them as regards their legal rights and codes of conduct, especially in questions of occupational health and safety. The firm encouraged employees to supervise compliance with the codes of conduct, and, more importantly, launched initiatives for employee representation through the creation of unions and works councils. Such actions foster communication with the community through worker-focused actions, encouraging community participation and empowerment, and helping enhance the recognition of companies.

## 7. Conclusions

Law 21/1991 on Prior Consultations in the Colombian Caribbean acts as a support for firm legitimacy as agents engaged in economic, social, and cultural development, safeguarding the protection of the areas in which they operate. Our logistic regression analysis reveals that the recognition or perception of the companies as agents of change in the community impacts the success of their CSR actions implemented to improve the conditions of the indigenous communities in Cartagena de Indias, as regards education, health, culture, employment, leadership, and other aspects related to economic, social, and cultural development, and the communities' recognition of the firms.

The greatest efforts made by the companies and agreed upon in the social licenses focus on education, job training, employment, and entrepreneurship for income generation, given that poverty can only be overcome through the capacities and opportunities provided by decent work. Additionally, the perceived benefits generated confirm that the process of economic inclusion in the communities, the improvement of their living standards, the

appreciation of their culture and ethnicity, and the appropriate and inclusive use of the environmental wealth of their territories are triggers for the communities' recognition of the companies as agents of change or recognized operators.

The territorial development approach used in these cases is guided by a model of intervention widely used by mining and energy companies in Latin America, pioneers in these types of processes. The model comprises three components: A productive and economic component supported by income generation; a social component with a focus on community infrastructures, public services, access to healthcare, education, proper nutrition, and transport; and a community agency component supported by the community's organizational capacity to manage its own development through participatory leadership.

The success of the company as an agent thus depends on the trust it generates in the community, the independence and appropriation of the processes by members of the community, the consolidation of the social fabric, which is implemented through communication and dialogue, participation, engagement and the focus of the local development formulated by the community. The conclusive relationships revealed by the logistic regression model confirm that the processes to recognize the integration of the company as an actor in the territory should always begin by opening channels of communication, dialogue, participation, and negotiation of differences. Only in this way can the company initiate the planning, implementation, and execution of the issues prioritized in their CSR.

In relation to CSR management, communities value investment in training and education from primary school to vocational training, practices that generate income. They also value respect for their culture, race, customs, and environmental wealth. The impacts of CSR and the recognition of the company as an agent forge the reputation of the company as regards its management and interrelationships with the community.

Given that healthcare is seen as a universal, fundamental right, with care, diagnosis, and treatment regarded as the responsibility of the State, CSR actions in healthcare tend to fall short, focusing on promotion and prevention. In addition, there is little impact on the relationship between suppliers and companies and the strengthening of community leadership, as they are seen as processes within the community that form part of their autonomy, preparation, and community cohesion.

Finally, although the four cases chosen for our research represent a clear representation of compliance with Law 21/1991 on Prior Consultation, we consider that one limitation of the research is tht it only covers four large corporations, despite their influence in the eight communities analyzed. At the same time, convenience sampling in ethnic contexts means that the results obtained may be biased toward the characteristics and experiences of the participants available. To mitigate these biases, we intend to implement complementary approaches, in future lines of research, consisting of the application of qualitative methods that help to enrich the understanding of ethnic and cultural diversity.

**Author Contributions:** Conceptualization, M.C.B.-C., Á.G.-M. and R.P.-M.; methodology, M.C.B.-C. software, M.C.B.-C.; validation, R.P.-M., Á.G.-M. and M.C.B.-C.; formal analysis, R.P.-M. and Á.G.-M.; investigation, M.C.B.-C.; resources, R.P.-M., Á.G.-M. and M.C.B.-C.; data curation, R.P.-M., Á.G.-M. and M.C.B.-C.; writing—original draft preparation, R.P.-M., Á.G.-M. and M.C.B.-C.; writing—review and editing, R.P.-M. and Á.G.-M.; visualization, R.P.-M. and Á.G.-M.; supervision, R.P.-M. and Á.G.-M.; project administration, R.P.-M. and Á.G.-M.; funding acquisition, R.P.-M. and Á.G.-M. All authors have read and agreed to the published version of the manuscript.

**Funding:** The publication and dissemination of the results of this article has received funding from the European Union through the European Rural Development Fund (FEDER).

**Institutional Review Board Statement:** Not applicable.

**Informed Consent Statement:** Not applicable.

**Data Availability Statement:** Data are not available at the moment because they are being used in other research.

**Conflicts of Interest:** The authors declare no conflict of interest.

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
