# Peer review of "Corporate Social Responsibility and Community Legitimacy: Colombian Caribbean Insights"

_sustainability, doi:10.3390/su151813659_

Round 1

Reviewer 1 Report

Dear authors 

The article looks interesting in the field of CSR. However, in the methodology part, you may need to add more information about the firms and communities. The importance of them and how they suppose to be engaged. Readers are not familiar with the current situation there, so they may need to have more information as your study is really dependent on the case study. You checked many hypotheses in your study. Why did you decide to include all of them, and why not any others? 

There are many errors in the reference list that you need to fix them. The title of the article looks very long. Meanwhile, you have only 4 keywords. I feel we need a better connection between these 2 parts. 

Author Response

Dear reviewer

We proceed to respond to each of your comments point by point. Within the document corresponding to the paper the changes derived from your suggestions are marked in green colour:

  1. The article looks interesting in the field of CSR. However, in the methodology part, you may need to add more information about the firms and communities. The importance of them and how they supposed to be engaged. Readers are not familiar with the current situation there, so they may need to have more information as your study is dependent on the case study.

In the Methodology section, we have added (page 15 and 16) a further explanation of the companies analysed. Also, on page 17 we have included a brief paragraph on the affected communities. The way in which companies must engage with communities is regulated under the umbrella of Law 21/91 on Prior Consultation. Reference is made to it throughout the work.

The added paragraphs are as follows:

Pages 15-16:

“…….

Serena del Mar is being built on the north coast of Cartagena, a new city designed by world leaders in urban planning and landscaping, which is based on the magical enclave of the Colombian Caribbean landscape. Serena del Mar is a bold proposal that takes a new look at the urban planning needs of the region, preserving and highlighting the natural diversity that enriches its surroundings. The new city incorporates global innovation in balance with the essence of a deep-rooted culture and breathtaking geographic beauty, giving the region new opportunities in housing, health, education, entertainment, hospitality and business. It is designed with citizens in mind, with a vision that maintains a balance between progress and roots, between tradition and the avant-garde. In the residential aspect, it will have a total of 17,000 homes for a population between 50,000 and 60,000 people. In recreation and infrastructure, it will have public areas for culture, entertainment and recreation, social clubs, marinas, canals, beaches, bike paths and trails with more than 100 kilometres. Institutional infrastructure will include a 70,000 m2 hospital, schools and a transportation terminal for the city, to be built in stages over the next few years.

Puerto Bahía is a multipurpose port complex located on a 155-hectare lot on Barú Island in the Bay of Cartagena. The port complex will take advantage of the robust prospect of crude oil production in Colombia and the need for more and better facilities for its transportation and export. The value proposition presented by Puerto Bahía is not limited to its design. The port has a strategic location in the Bay of Cartagena, itself one of the largest trade centres in Latin America. Puerto Bahia's competitive advantages are evident in its marketing strategy. The port has an initial storage capacity for liquids of 3.3 million barrels, a general cargo handling facility for transhipment, dry cargo, and general bulk cargo. The port is optimally designed to meet current demand, but also has sufficient space available for future expansion.

Karibana is a one-of-a-kind development with a wide range of entertainment and leisure activities. This project reflects the brand's intention to be present in each of the world's most important gateway cities and most sought-after leisure destinations. The project is located north of the city of Cartagena, between the towns of Manzanillo and Punta Canoa. In addition to representing a complete life experience, surrounded by the best facilities for the rest and peace of families in a natural and culturally rich environment, since 2008 it is the project with the highest value in Cartagena. The investment for this project will be 150 million pesos, which is expected to be inaugurated in December 2016. Surtigas is a company with 43 years of experience, during which it has provided access to natural gas and associated services in Colombia to more than 550,000 households. The company currently operates in more than 120 towns in the departments of Bolívar, Sucre, Córdoba, Antioquia and Magdalena, making it the natural gas distributor and marketer with the largest geographic area served in the national territory: 90,000 km2. In these four decades, Surtigas has achieved 90% coverage in its area of influence, bringing energy solutions and well-being to thousands of families, 80% of whom belong to strata 1 and 2. It also provides solutions to 5,531 commercial customers and more than 315 industrial customers.

With a CSR management system focused on sustainability, Surtigas aims to promote more equitable, liveable and viable scenarios in the country through a balance between the social, economic and environmental aspects of its business actions [99]”.

Page 17:

“……These communities are made up of Afro-descendant and Raizal ethnic minorities, which are the most representative in the Region. This has served to guarantee the participation of these ethnic minorities in the protection of their ethnic and cultural integrity before initiating projects to search for and exploit natural resources in their territories”.

  1. You checked many hypotheses in your study. Why did you decide to include all of them, and why not any others? 

All the hypotheses have been formulated because they all involve specific actions that Law 21/91 on Prior Consultation establishes as commitments of the companies, in their exercise of Corporate Social Responsibility, towards the communities. Among these actions are:

  1. Health Improvement: collaboration in community health programs; health infrastructure support: Contribute to the development and maintenance of local health centres, clinics and hospitals to improve access to health care or health education: Provide educational resources on health and wellness to empower communities to make informed decisions about their health.
  2. Educational Programs: collaborate with local educational institutions to develop educational programs that promote leadership and skills training for community members.
  3. Income Enhancement: local employment generation.
  4. Training and skills development: offer education and training programs that improve the skills and competencies of community members, which can lead to better employment opportunities or local Entrepreneurship Support.
  5. Environmental Improvement: environmental sustainability; conservation projects or environmental Education.
  6. Support for Culture: cultural Sponsorship; promotion of local arts and crafts or cultural workshops and programs.
  7. Supply Chain Support: prioritize the procurement of goods and services from local suppliers, which can boost the community's economy and create jobs or provide training and advice to local suppliers to improve their business practices and meet quality and sustainability standards.
  8. Leadership Development: leadership development programs, mentoring and mentoring or scholarships and Grants: Provide scholarships and grants to young community leaders to support their education and leadership training.

These actions are explained and supported by the existing literature in section 3 and its subsections.

  1. There are many errors in the reference list that you need to fix them.

All references have been reviewed, corrected, updated and put in the format required by the MDPI. In blue colour you will find the new references that have arisen, due to the update of the bibliography related to the research from 2019 to 2023. These references have also been considered in the Discussion section.

  1. The title of the article looks very long. Meanwhile, you have only 4 keywords. I feel we need a better connection between these 2 parts. 

We propose another shorter title as well as a new list of Key words so that, if the editor considers it convenient, the change can be made. We agree with your comment.

FULL TITLE:

Corporate Social Responsibility and Community Legitimacy. Colombian Caribbean Insights.

KEY WORDS

Business Ethics, Environmental Sustainability, Local Communities, Indigenous Rights, Corporate Accountability, Public Participation. Socioeconomic Development, Legal Framework, Stakeholder Collaboration. Community Empowerment.

Thank you very much for your comments, we consider them to be very valuable and they will undoubtedly contribute to the improvement of this research.

Best regards,

The authors

Reviewer 2 Report

This article explores the factors that influence the recognition and legitimacy of companies by communities. The item is really interesting. However, there are several weak points that should be addressed before the article can be accepted. To improve the article and address the weak points, the following revisions can be made:

 Update the bibliography:

The authors should supplement the bibliography with recent articles and research to ensure that the information and references used are up-to-date. This is especially important in the discussion section, where they should compare their findings with recent studies to validate their conclusions.

Properly Format Bibliography and References:

The authors need to ensure that the bibliography and references are properly written according to the standard specified by the MDPI guidelines. Consistency and accuracy in citation formatting are crucial for the credibility of the article.

Structure of the paper

The article's structure will be more organized and reader-friendly, allowing readers to better understand the rationale behind each hypothesis and its connection to the existing literature.

1.      Organize the Literature review into Sections:

Group the Literature review into smaller sections with appropriate titles that correspond to the main topics or hypotheses discussed in the article.

2.      Mention Relevant Articles in Each Section:

Within each section, list the relevant articles that support or are related to the specific hypothesis being discussed. Include a brief summary of key findings of each article to highlight its relevance to the hypothesis.

3.      Formulate Corresponding Hypotheses:

At the end of each section, formulate the corresponding hypothesis based on the collective findings and evidence presented in the listed articles.

Clarify the Sampling Methodology:

 The authors mention: “The sample comprised 98 respondents, with the distribution across the communities surveyed reflected in Table 1, with a sample error of 9.8% within a 95% confidence interval". The authors should provide a clear explanation of the sampling method employed in the research. Whether it was random sampling, systematic sampling, or another method, it needs to be explicitly stated. Furthermore, they should describe how the selection was made to ensure transparency and reproducibility of the study.

Justify Sample Size and Generalizability:

The authors should address the issue of sample size in the research limitations section. They should provide a justification for why the sample size of 98 respondents was considered adequate and representative. If possible, they can cite statistical principles or prior research that supports the chosen sample size. In the results section, the authors comment on their findings to the community. In many cases, this means that they refer to 2 or 3 companies. Therefore, the authors should acknowledge the limitations of generalizing results based on such  small samples and offer suggestions for future research that can address this limitation.

Data Analysis

The authors conducted regression analysis in the study; however, the equations and their explanations were not provided. It would be beneficial to include the equations and explanations to enhance the understanding of the analysis. Moreover, considering the complexity of the data and the presence of multiple variables, it may be worth considering multiple regression analysis. Multiple regression allows to examine the relationship between multiple independent variables and a single dependent variable simultaneously, which can provide more comprehensive insights and potentially improve the model's predictive capabilities

Author Response

Dear reviewer

We proceed to respond to each of your comments point by point. Within the document corresponding to the paper the changes derived from your suggestions are marked in blue colour:

This article explores the factors that influence the recognition and legitimacy of companies by communities. The item is interesting. However, there are several weak points that should be addressed before the article can be accepted. To improve the article and address the weak points, the following revisions can be made:

  1. 1Update the bibliography:

The authors should supplement the bibliography with recent articles and research to ensure that the information and references used are up to date. This is especially important in the discussion section, where they should compare their findings with recent studies to validate their conclusions.

All references have been reviewed and corrected. In blue colour you will find the new references that have arisen, due to the update of the bibliography related to the research from 2019 to 2023. These references have also been considered in the Discussion section.

  1. Properly Format Bibliography and References:

The authors need to ensure that the bibliography and references are properly written according to the standard specified by the MDPI guidelines. Consistency and accuracy in citation formatting are crucial for the credibility of the article.

All references have been put in the format required by the MDPI.

  1. Structure of the paper

The article's structure will be more organized and reader-friendly, allowing readers to better understand the rationale behind each hypothesis and its connection to the existing literature.

3.1. Organize the Literature review into Sections:

Group the Literature review into smaller sections with appropriate titles that correspond to the main topics or hypotheses discussed in the article.

3.2. Mention Relevant Articles in Each Section:

Within each section, list the relevant articles that support or are related to the specific hypothesis being discussed. Include a summary of key findings of each article to highlight its relevance to the hypothesis.

3.3. Formulate Corresponding Hypotheses:

At the end of each section, formulate the corresponding hypothesis based on the collective findings and evidence presented in the listed articles.

As you can see, section 3, corresponding to the literature review and hypothesis proposal, has been almost completely reworked. Specific subsections (3.1 to 3.8) have been established in which the most relevant bibliography to support each hypothesis has been included. In each subsection, the main idea discussed in the most important literature is provided. In addition, at the end of each subsection the corresponding hypothesis is formulated.

  1. Clarify the Sampling Methodology:

 The authors mention: “The sample comprised 98 respondents, with the distribution across the communities surveyed reflected in Table 1, with a sample error of 9.8% within a 95% confidence interval". The authors should provide a clear explanation of the sampling method employed in the research. Whether it was random sampling, systematic sampling, or another method, it needs to be explicitly stated. Furthermore, they should describe how the selection was made to ensure transparency and reproducibility of the study.

A paragraph has been included in the Methodology section (pages 16 and 17) to clarify the sampling method used in the research.

The paragraph is as follows:

“…..To configure the sample and therefore the set of people to whom the questionnaire was addressed, the person interviewed was selected based on his or her representativeness in the group of actors involved. By representativeness we mean the capacity of certain people to embody the feelings and thoughts of the social segment to which they belong, so that the researcher should select the person who offers an important opportunity for learning and providing information [100].

In this case, the interview was conducted with each main CSR manager with the prioritised ethnic minority communities, with good communication and fluency of content, achieving the joint construction of the meanings of the themes. A short, semi-structured questionnaire was used to collect all the impressions and perceptions of the manager, adding all the supporting documents provided by the company”.

  1. Justify Sample Size and Generalizability:

The authors should address the issue of sample size in the research limitations section. They should provide a justification for why the sample size of 98 respondents was considered adequate and representative. If possible, they can cite statistical principles or prior research that supports the chosen sample size. In the results section, the authors comment on their findings to the community. In many cases, this means that they refer to 2 or 3 companies. Therefore, the authors should acknowledge the limitations of generalizing results based on such small samples and offer suggestions for future research that can address this limitation.

In the Methodology section we have justified why we consider the sample size to be adequate. to this end, we have incorporated footnote 2.

Also in the Conclusions section, we have included a paragraph indicating that both the sample size and the number of companies analysed may be a limitation of the research and that in future lines we intend to extend these variables.

The paragraph is as follows:

“….

Finally, although the four cases chosen for our research represent a clear representation of compliance with the Law 21/1991 on Prior Consultation, we consider that one limitation of the research is since it only covers four large corporations, despite their influence in the eight communities analysed.  At the same time, the answers obtained in the interviews are considered another limitation of the survey, and therefore, the need to expand the sample is considered as a future line of research”.

  1. Data Analysis

The authors conducted regression analysis in the study; however, the equations and their explanations were not provided. It would be beneficial to include the equations and explanations to enhance the understanding of the analysis. Moreover, considering the complexity of the data and the presence of multiple variables, it may be worth considering multiple regression analysis. Multiple regression allows to examine the relationship between multiple independent variables and a single dependent variable simultaneously, which can provide more comprehensive insights and potentially improve the model's predictive capabilities.

In table 2 of the Methodology section, we have coded the variables (dependent and independent) and at the end of this section we have formulated the equation that summarises the logistic regression model applied.

“… The equation below represents the model used which represents the dependent variable as a function of all the independent variables used and coded as shown in table 2.

Regarding your comment on the choice of model, the authors understand that the choice of model is determined by the nature of the variables involved and the type of question you are trying to answer. If the simple reply variable is binary and you are interested in the probability of success, we consider binary logistic regression to be more appropriate. If you are working with a continuous reply variable and want to understand how several predictor variables influence it in a linear fashion, multiple regression would be more appropriate in this case.

Following this consideration, we have opted for a binary logistic regression model. This is reflected in the Methodology section (page 19) in this sentence:

“…The choice between binary logistic regression has been determined by the nature of the variables involved and the type of question you are trying to answer”.

Thank you very much for your comments, we consider them to be very valuable and they will undoubtedly contribute to the improvement of this research.

Best regards,

The authors

Round 2

Reviewer 2 Report

You have responded to the majority of my comments. Well done.

However, concern persists within the methodology section. The assertion made in the paper states, "The sample comprised 98 respondents, drawn from a total population of 57,800 inhabitants across 557 populations. The distribution of these respondents across the surveyed communities is presented in Table 1, accompanied by a sample error of 9.8%2 within a 95% confidence interval."

 While attempting to assess the sampling error, it's crucial to employ a method that ensures the sample's accuracy in mirroring the population's characteristics. Random or systematic sampling techniques are generally employed for this purpose. You seem to suggest that the individuals interviewed were selected based on their representativeness within the group of actors involved, yet the mechanism by which this representation was ensured remains unclear. It's worth considering whether the methodology effectively supports this claim.

 Upon closer examination, I think that the sampling approach used may lean towards a convenience sample, rather than through a randomized process. If this indeed is the case, it would be inappropriate to calculate a sampling error in the traditional sense. Instead, the paper could potentially delve into discussions about the presence of sample bias, exploring how the method of participant selection could impact the generalizability of the findings. In this point, you have to reinforce the methodology with relevant literature that either substantiates their method of sampling as representative or underscores the limitations associated with the sampling method.

Author Response

Revisor 2. Round 2

Dear reviewer we have responded to the consideration made by replying in the manuscript in red colour.

However, concern persists within the methodology section. The assertion made in the paper states, "The sample comprised 98 respondents, drawn from a total population of 57,800 inhabitants across 557 populations. The distribution of these respondents across the surveyed communities is presented in Table 1, accompanied by a sample error of 9.8%2 within a 95% confidence interval."

 While attempting to assess the sampling error, it's crucial to employ a method that ensures the sample's accuracy in mirroring the population's characteristics. Random or systematic sampling techniques are generally employed for this purpose. You seem to suggest that the individuals interviewed were selected based on their representativeness within the group of actors involved, yet the mechanism by which this representation was ensured remains unclear. It's worth considering whether the methodology effectively supports this claim.

Upon closer examination, I think that the sampling approach used may lean towards a convenience sample, rather than through a randomized process. If this indeed is the case, it would be inappropriate to calculate a sampling error in the traditional sense. Instead, the paper could potentially delve into discussions about the presence of sample bias, exploring how the method of participant selection could impact the generalizability of the findings. In this point, you must reinforce the methodology with relevant literature that either substantiates their method of sampling as representative or underscores the limitations associated with the sampling method.

We are very grateful to the reviewer for his appreciation, which helps us to correct an inadequate explanation of the sampling method applied. In fact, the sampling method was a non-probabilistic convenience sampling.

The main reason why we have chosen to use this sampling method is justified by its traditional use in ethnic contexts. This is because, on occasions, accessibility to certain ethnic groups is limited due to cultural, linguistic, political, or geographic barriers. Thus, in research on specific cultural traditions and practices of ethnic minority or indigenous groups, selection by convenience may be justifiable due to the limited availability of participants who meet the inclusion criteria.

There is a considerable literature on this subject. In the manuscript we have pointed out some of these more recent investigations where this sampling method is used in the study of ethnicities or indigenous populations. Specifically, Mora et al. (2023), Moreno et al. (2022), and Echevarría et al. (2020).

These references have been added in the bibliography section:

  • Otzen, T., & Manterola, C. (2017). Sampling techniques on a population study. International Journal of Morphology, vol. 35 no. 1 (2017) p. 227-232.
  • Mora, G., Ruiz-Díaz, L., Aquino, V., & Ayala Ayo, J. F. (2023). Deshumanización Evidente de una Muestra de Psicólogos Paraguayos hacia sus Pacientes. Revista Científica de la UCSA10(1), 13-18.
  • Echeverria-López, S., Henríquez-D’Aquino, E., Werlinger-Cruces, F., Villarroel-Díaz, T., & Lanas-Soza, M. (2020). Determinantes de caries temprana de la infancia en niños en riesgo social. International journal of interdisciplinary dentistry13(1), 26-29.
  • Moreno-Acero, I. D., Velásquez, M. L. S., & Vargas, A. M. C. (2022). Desterritorialización y transformación de las dinámicas cotidianas de las familias víctimas del conflicto armado colombiano. Collectivus, Revista de Ciencias Sociales9(2), 175-232.

In response to your comments, we have included the following paragraph in the methodology section:

“To configure the sample, a non-probabilistic convenience sampling method has been applied [100], where the inclusion criterion has been the evidence of the representativeness of the respondent in the group of actors involved [101, 102 and 103]. By representativeness we mean the capacity of certain people to embody the feelings and thoughts of the social segment to which they belong, so that the researcher should select the person who offers an important opportunity for learning and providing information [104].

In this case, the interview was conducted with each main CSR manager with the prioritised ethnic minority communities, with good communication and fluency of content, achieving the joint construction of the meanings of the themes. A short, semi-structured questionnaire was used to collect all the impressions and perceptions of the manager, adding all the supporting documents provided by the company. The questionnaire was thus the primary data source and was aimed at the members of the local community involved in the business projects selected. This questionnaire was completed by all the CSR managers of the eight ethnic communities. Their responses resulted in a sample of 98 interviews distributed as shown in Table 1”.

Also, in the conclusions section we have reflected the limitation of using this sampling method:

“At the same time, convenience sampling in ethnic contexts means that the results obtained may be biased towards the characteristics and experiences of the participants available. To mitigate these biases, we intend to implement complementary approaches, in future lines of research, consisting of the application of qualitative methods that help to enrich the understanding of ethnic and cultural diversity”.

We sincerely thank you for the comment you have made.

Best regards,

The authors

Round 3

Reviewer 2 Report

Thank you for addressing my comments regarding the sampling method.

Author Response

Revisor 2. Round 3

Dear reviewer we respond to your comment in this third round which does not entail any changes to the latest version of the manuscript.

Reviewer's comment:

Thank you for addressing my comments regarding the sampling method.

Reply:

We would like to thank you for your contributions as we believe they have made an invaluable contribution to the improvement of the research.

The attached version of the manuscript contains no changes to the last submission.

We sincerely thank you for the comment you have made.

Best regards,

The authors
